# Hybrid Machine Learning Approach to Zero-Inflated Data Improves Accuracy of Dengue Prediction

**Micanaldo Ernesto Francisco[1,2,3], Thaddeus M. Carvajal[1,4], Kozo Watanabe** **[1] \***

**1** Center for Marine Environmental Studies (CMES), Ehime University, Matsuyama, Japan, **2** Graduate School of Science and Engineering, Ehime University, Matsuyama, Ehime, Japan, **3** Faculty of Architecture and Physical Planning (FAPF), Lurio University, Nampula, Mozambique, **4** Department of Biology De La Salle University, Taft Ave Manila, Philippines

\* watanabe.kozo.mj@ehime-u.ac.jp

## Abstract

### Background

Spatiotemporal dengue forecasting using machine learning (ML) can contribute to the development of prevention and control strategies for impending dengue outbreaks. However, training data for dengue incidence may be inflated with frequent zero values because of the rarity of cases, which lowers the prediction accuracy. This study aimed to understand the influence of spatiotemporal resolutions of data on the accuracy of dengue incidence prediction using ML models, to understand how the influence of spatiotemporal resolution differs between quantitative and qualitative predictions of dengue incidence, and to improve the accuracy of dengue incidence prediction with zero-inflated data.

### Methodology

We predicted dengue incidence at six spatiotemporal resolutions and compared their prediction accuracy. Six ML algorithms were compared: generalized additive models, random forests, conditional inference forest, artificial neural networks, support vector machines and regression, and extreme gradient boosting. Data from 2009 to 2012 were used for training, and data from 2013 were used for model validation with quantitative and qualitative dengue variables. To address the inaccuracy in the quantitative prediction of dengue incidence due to zero-inflated data at fine spatiotemporal scales, we developed a hybrid approach in which the second-stage quantitative prediction is performed only when/where the first-stage qualitative model predicts the occurrence of dengue cases.

### Principal findings

At higher resolutions, the dengue incidence data were zero-inflated, which was insufficient for quantitative pattern extraction of relationships between dengue incidence and environmental variables by ML. Qualitative models, used as binary variables, eased the effect of data distribution. Our novel hybrid approach of combining qualitative and quantitative

**Data Availability Statement:** The data underlying the results presented in the study are available at the following link: https://doi.org/10.5061/dryad. x3ffbg7ss

**Funding:** This study was financially supported by the Japan Society for the Promotion of Science (JSPS) Grant-in-Aid Fund for the Promotion of Joint International Research (Fostering Joint International Research (B)) under grant number 19KK0107, JSPS Grant-in-Aid for Scientific Research (A) under grant number 19H01144, the JSPS Core-to-Core Program B. Asia-Africa Science Platforms, and the Endowed Chair Program of the Sumitomo Electric Industries Group Corporate Social Responsibility Foundation to KW. The funders had no role in study design, data collection and analysis, decision to publish, or preparation of the manuscript.

**Competing interests:** The authors have declared that no competing interests exist.

predictions demonstrated high potential for predicting zero-inflated or rare phenomena, such as dengue.

## Significance

Our research contributes valuable insights to the field of spatiotemporal dengue prediction and provides an alternative solution to enhance prediction accuracy in zero-inflated data where hurdle or zero-inflated models cannot be applied.

## Author summary

In our study, we tackled the complex challenge of predicting dengue fever outbreaks, a crucial task in the field of epidemiology. Dengue prediction is complicated because it relies on the quality of data, which may be affected by the temporal and spatial resolution. We explored different machine learning algorithms across various spatial (village, city and region) and temporal resolutions (weekly and monthly). A key hurdle we encountered was the high frequency of zero values in reported dengue cases, a common issue known as zero-inflated data. This phenomenon makes accurate predictions difficult, especially at finer resolutions. To overcome this obstacle, we first made qualitative predictions about the presence or absence of dengue cases. Then, in scenarios indicating disease presence, we estimated the magnitude of cases quantitatively. This innovative method we designated as hybrid approach and significantly enhanced prediction accuracy in zero-inflated data. This approach can be applied to continuous data where zero-inflated or hurdle models cannot be applied. Our findings have broader implications beyond dengue prediction, shedding light on the challenges of dealing with zero-inflated data in various real-world situations. By improving our understanding of these complexities, our research contributes valuable insights that not only benefit scientists working in epidemiology but also have practical applications in public health strategies ensuring more effective and targeted interventions.

## Introduction

Machine learning (ML)-based approaches have become popular in the field of environmental epidemiology since they can forecast the risk of infectious diseases, such as dengue fever on different spatial or temporal scales based on environmental variables. ML approaches have been largely adopted for forecasting because of their efficiency in handling different types of datasets, including outliers, higher dimensionality, and even complicated nonlinear interactions among variables [1]. Climate data (e.g., temperature, precipitation, wind speed, and relative humidity) [2] and land use types are environmental variables often used as predictors of dengue because they are important determinants of vector mosquito ecology, including reproduction, longevity, biting rate, virus replication, and transmission ability [3].

Given the limited spatial coverage of weather stations (WSs), recent studies have preferred remotely sensed climatic data over meteorological data obtained from ground-based WSs. The advantages of remote sensing (RS) include the ability to regularly collect information over large spatial and inaccessible areas and monitor their changes over time [4]. In spatial forecasting, having both predictors and dependent variables with the same spatial resolution is necessary to make analyses viable [5]. Most dengue cases have been reported based on the

administrative units (e.g., village, city, or province) in many countries. To match the spatial resolution of dengue incidence with RS data, many studies have adopted these administrative units as the spatial units for analysis [6,7,8]. Consequently, the data were notably influenced by the relationship between pixel size and administrative unit. For instance, if the pixel size of RS data is smaller than the dimensions of the administrative units, spatial variability within these units for specific RS variables is likely to occur. Conversely, if the pixel size of RS data exceeds the size of the administrative units, there may be limited to no variation between administrative units for a given variable. Therefore, modifying the pixel format of RS data into administrative boundaries may generate different results depending on the scale ratio between the RS pixel size and the size of administrative units [9].

Although dengue incidence predictions at finer spatial and temporal resolutions would allow for more efficient and targeted preventive measures (e.g., vector control), accurate prediction at fine resolutions is more difficult [10]. We should explore the optimal resolutions that allow for accurate predictions and are useful for dengue control, which is a tradeoff relationship. Although many studies have used RS data for dengue forecasting at the national [11], regional [2], city [12], and village [13] scales, the implications of using RS data on model accuracy with different spatial resolutions have not been investigated. Furthermore, although several ML algorithms have been used in dengue forecasting, most studies have focused on evaluating and comparing the prediction accuracy of different algorithms at one specific spatial or temporal resolution [13,14]. However, the optimal algorithm for each spatial or temporal resolution of data and type of response variable (quantitative or qualitative) may vary from case to case. For example, although the random forest was reported to outperform many algorithms [15,2], other algorithms, such as the generalized additive model [16] and support vector machines and regression [13,17], were also found to perform better in other studies. Further studies with comparable datasets are required to evaluate the performance of different ML algorithms.

Although qualitative models aim to provide a simple answer in which villages dengue cases are likely to occur, quantitative models can estimate the magnitude of cases in each village. The latter may provide more useful information for developing preventive measures. Although quantitative dengue predictions at large spatial scales are relatively accurate [12,11,18,17], accurate quantitative predictions at fine scales are difficult because of a lack of accuracy in dengue data reporting [3] or zero-inflated data, which occur when a significant portion of the data contain zero values. This can lead to challenges in modeling and prediction because the excessive number of zero values may not necessarily reflect the true absence of dengue cases but rather an underreporting or underdetection of cases in certain areas or periods [19]. Consequently, accurately capturing the dynamics of dengue at fine scales demands sophisticated statistical techniques that can effectively handle such data distributions.

Generating ML models in the standard fashion to predict zero-inflated data may generate biased predictions towards the non-zero values. While zero-inflated [20] and hurdle [21] models were proposed to address such data, their primary design is for count data outcomes with discrete distribution (i.e., Poisson or negative binomial), which may not suit continuous data with non-integer values. The hurdle model first employs a binary model to predict zero or positive responses and second, a truncated model for positive data. All zeros are generated from the binary model while the non-zero values by the truncated component. In contrast, zero-inflated models allow zeros to be produced by both the binary and count components. Their two-step approach is robust in improving model accuracy [22]. However, the application of these models to continuous data typically following a Gaussian distribution is not possible due to a mismatch with discrete data assumptions. Alternative approaches have combined ML with upsampling or downsampling techniques to improve the performance of predictive

models. While these methods are commonly employed in addressing imbalanced data with discrete outcomes [23], their applicability to continuous data is constrained by their limited ability to improve model accuracy. For predictive modeling with continuous data, alternative approaches are necessary to ensure robust and reliable predictions. Combining the two-step concept of hurdle models with the predictive capabilities of ML methods is an avenue worth exploring in handling zero-inflated data.

This study investigated new strategies to improve dengue prediction accuracy, focusing on the effects of spatial and temporal resolutions and the phenomenon of zero-inflation in continuous data, which have not been deeply considered in environmental epidemiology modeling. Six ML algorithms were used to perform qualitative and quantitative dengue predictions in Metropolitan Manila, Philippines, and their prediction accuracy was compared at different spatial (village, city, and region) and temporal (weekly and monthly) resolutions. Subsequently, to improve the accuracy of quantitative dengue forecasting and address the issue of zero-inflation data commonly encountered in infectious disease risk forecasting, we developed a hybrid approach in which the second-stage quantitative forecasting is performed only when/ where the first-stage qualitative model predicts the occurrence of dengue cases.

## Materials and methods

### Study area

Metropolitan Manila is the National Capital Region of the Philippines, located in Southwestern Luzon (14˚ N Latitude, 121˚E Longitude) with an area of 636 $km^2$ and 100% urbanization [24]. It comprises 16 cities and one municipality (hereafter "region") with a total population of 13,484,462 [25]. Each city or municipality (hereafter "city") is subdivided into barangays, commonly known as villages (n = 1,706), which are the smallest administrative division. Small villages were merged into a zone in the cities of Manila, Caloocan, and Pasay based on Francisco, et al. (2021) [26]. Afterwards, a total of 27 villages containing incomplete reports were removed from the final dataset. Therefore, 437 villages or zones (hereafter "villages") were used in subsequent analyses in this study. Population statistics were obtained from the Philippine Statistics Authority agency (www.psa.gov.ph). Because the Philippine population census is conducted every 5 years, we obtained the 2010 [27] and 2015 [28] census data and used the compounded population growth rate to calculate the population for the years 2009, 2011, 2012, and 2013 for each village.

According to the Health Surveillance and Informatics Division, National Epidemiology Center of the Department of Health, every year, thousands of dengue cases have been reported in the country. The annual number of dengue cases in the country increased from 21,794 cases in 2009 [29] to more than 113,000 cases in 2014 [30]. Metropolitan Manila always ranks among the top three most affected regions every year with 4,259 cases in 2009 [29] and 23,437 cases in 2014 [30].

### Epidemiological data

The total numbers of weekly reported dengue cases from January 2009 to December 2013 for each of the 437 villages were obtained from the National Epidemiology Center, Department of Health, Philippines. Most reported dengue cases during this period were suspected or probable cases according to standard definitions and were not confirmed in a laboratory. Dengue incidence in Metropolitan Manila exhibited consistent seasonal peaks following the rainy season. We calculated the weekly dengue incidence rate by dividing the total number of dengue cases each week by the total population of the village multiplied by a population factor of 10,000. We chose to model dengue data as incidence rates to account for variability in the population size

across different geographic areas. This approach allows for a more nuanced understanding of the data, facilitating comparisons between areas with different population sizes. The Dengue incidence rate was converted to the natural logarithm [log (n + 1)] after adding 1 to the value to address the skewness in the data (S1 Fig).

### Satellite remotely sensed data

We obtained Global Precipitation Measurement (GPM) Integrated Multi-Satellite Retrievals for GPM (IMERG) V6 precipitation data with hourly temporal resolution and 1-degree spatial resolution in the "precipitationCal" band from the "NASA/GPM_L3/IMERG_V06" collection [31]. The IMERG is the GPM's Level 3 multi-satellite precipitation algorithm that combines intermittent precipitation estimates from all constellation microwave sensors, infrared-based observations from geosynchronous satellites, and gauge precipitation data [32].

The moderate resolution image spectroradiometer (MODIS) sensor onboard the Terra and Aqua satellites collected the average land surface temperature (LST) and normalized difference of vegetation index (NDVI). For LST, 1-km spatial resolution data from the MOD11A1 and MYD11A1 products were used [33,34]. For the NDVI, 250-m spatial resolution data from MOD13Q1 and MYD13Q1 were used [35]. Additional processing to fill missing pixels in both LST and NDVI raster datasets was performed using the locally weighted regression (LWR) method in GRASS GIS, version 7.8.3 [36](S1 Video). LWR can reconstruct spatiotemporal data and provide a fully gap-free dataset [37]. The reconstructed LST data were further validated with data at three ground WSs in Metro Manila (Ninoy Aquino International Airport WS: $r = 0.81$, $p < 0.001$; Manila WS: $r = 0.77$, $p < 0.001$; and Science Garden WS: $r = 0.83$, $p < 0.001$).

The ERA5-Land product of the European Centre for Medium-Range Weather Forecasts offers diverse climatic data reanalysis, including wind speed, dew point, and air temperature. Wind speed has two components, eastward and northward, consisting of hourly measures of horizontal wind speed at 10 m above the ground with 1 degree of spatial resolution [38]. Each wind component was used as independent time series data in the analysis. Because a dedicated relative humidity (RH) satellite dataset is not yet available, we estimated the RH by dividing the water vapor pressure by the saturation water vapor pressure in the atmosphere, multiplied by 100 [39]. Water vapor pressures were priorly calculated using the dew point and mean air temperature from ERA5-Land [38] data. The estimated RH was further validated using data from the three local ground WSs (Ninoy Aquino International Airport WS: $r = 0.85$, $p < 0.001$; Manila WS: $r = 0.94$, $p < 0.001$; and Science Garden WS: $r = 0.94$, $p < 0.001$).

All RS data (S1 Table and S2 Fig) were acquired using the Google Earth Engine (GEE) code editor platform [40]. The GEE code editor is a web-based integrated development environment for writing and running Java scripts to support geospatial analysis [41]. The integrated availability and accessibility of diverse satellite data, band filtering and exporting, cloud computation, and computational time and cost reduction are among the advantages of using GEE. Additional preprocessing of the satellite data was performed using GRASS GIS, version 7.8.3 [36]. The climatic data in raster format was averaged for each village considering the pixels that overlapped with the respective village boundaries. However, opting for the conventional mean value across all pixels was avoided, as it could introduce bias to the mean values. Instead, a spatially weighted average was employed. This entailed calculating the weighted average for a climatic variable within a village by summing the products of pixel values and the respective percentage of the village overlapped by each pixel. This approach ensures a fair and unbiased extraction of spatial data from pixels to village polygons.

### Land use, road, and flood risk data

The percentage of LU, road network density (RND), and weighted average of flood risk data per spatial unit (village, city, or region) were calculated according to Francisco, et al. (2021) [26]. The local land-use map of Metropolitan Manila (2004) was obtained from the Philippine Geoportal website (www.geoportal.gov.ph), which was managed by the Bureau of National Mapping and Resource Authority and the Metropolitan Manila Development Authority [42]. This map was further updated with open street maps and Geofabrik GmbH data (www.geofab-rik.de). Other datasets, such as RND, obtained from the Philippine GIS Data Clearinghouse website (http://philgis.org/) and the flood hazard map of Metropolitan Manila obtained from the Light Detection and Ranging LiDAR Portal for Archiving and Distribution website (https://lipad.dream.upd.edu.ph) [43] were also used as predictors. The data used in this study is deposited in a public repository [44].

### Cross-correlation analysis

A cross-correlation analysis using the entire data from 2009–2013 was performed between the weekly or monthly temporal variations in environmental factors (i.e., minimum temperature, mean temperature, maximum temperature, precipitation, northward wind, eastward wind, RH, and NDVI) and dengue incidence. This analysis aimed to uncover the lags at which these factors most strongly correlate with dengue incidence, which is essential for constructing a model with meaningful predictive features. While the entire data was used at this exploratory step, the actual predictive model did not have access to the 2013 data during training to avoid information leakage to the testing step. The temporal mean value per week or month for each village was used for each environmental factor. We identified the best-lag time based on the highest absolute value of Pearson correlation coefficient and whether the *p*-value is ≤0.05. Lagged features account for the time delays in the impact of climatic factors on dengue incidence, providing a more accurate representation of the underlying dynamics and improving model predictive power [45]. Thirty-two time points for weeks and seven time points for months were analyzed from lag zero (dengue incidence in the current week correlated to the environmental variable in the current week) to lag 31 (dengue incidence in the current week correlated to the environmental variable in the earlier week 31). The identified best-lag time results are presented in S2 Table. The input dataset for the dengue prediction models used the set of independent variables with the best-lag time that showed the highest correlation. All analyses were performed using R, version 4.1.0.

### Modeling algorithms and their implementation

Quantitative models were used to predict dengue incidence at six combinations of two temporal and three spatial resolutions (i.e., week-village, week-city, week-region, month-village, month-city, and month-region). Region refers to the entire administrative area of Metropolitan Manila, which is divided into 17 cities, which are then divided into 437 villages. In contrast, dengue occurrence was predicted only at the week-village resolution using qualitative models. This was because attempting to predict the presence or absence of dengue at larger spatial scales (i.e., city or region) or temporal scales (i.e., month) would not yield practical and actionable information for effective interventions. The reason for this limitation lies in the nature of the qualitative models. At finer scales, a binary prediction of the presence or absence of dengue can be instrumental for local health authorities in planning targeted interventions. However, at larger scales, a binary prediction for an entire city or region might not provide the detailed information required for effective decision-making and resource allocation. In constructing these dengue prediction models, we compared six ML algorithms—generalized additive

models (GAM), random forests (RF), condition inference forest (CIF), artificial neural networks (ANN), support vector machines and regression (SVM and SVR), and extreme gradient boosting (XGB). Epidemiological and environmental data from 2009 to 2012 were used for training, and data from 2013 were used for model validation. Quantitative models predicted dengue incidence, whereas qualitative models predicted the presence or absence of dengue cases. These models assumed static landscape features (percentage of land-use category, weighted average of flood risk, and RND) with the best temporal lagged mean values of weekly or monthly minimum temperature, mean temperature, maximum temperature, precipitation, RH, eastward wind speed, northward wind speed, and NDVI per village, city, or region as predictors (S3 Table). The dynamics in dengue incidence followed distinct patterns in each of 437 villages. To examine the differences in the predictability of temporal changes in dengue incidence within individual village, we developed weekly quantitative and qualitative models to predict the temporal changes for each individual village with CIF. These 437 models were developed using climatic variables only for the 260 weeks (2009–2013) because the LU data were static. Unlike the spatiotemporal models that cover diverse locations with varying land use and climatic patterns (spatial and temporal heterogeneity), the individual village models only account for temporal changes in climate. Therefore, excluding static features like land use is reasonable as it avoids unnecessary model complexity without losing explanatory power. We followed the same data-splitting strategy with 2009–2012 for train and 2013 for testing. The accuracy of the temporal model for each village was evaluated using $R^2$ (quantitative model) or AUC values (qualitative models). To determine the factors affecting the accuracy of the temporal predictions by village, we examined the correlation between spatial variations in $R^2$ or AUC values among the 437 villages and the spatial variations in the number of dengue cases, dengue incidence, frequency of dengue occurrence, area, population size, and spatial proportion of LU types per village.

The accuracy of the quantitative models was measured using the root-mean-square error (RMSE): R-squared ($R^2$) for temporal models and standardized adjusted $R^2$ (S-Ra$^2$) for spatiotemporal models. The S-Ra$^2$ metric was used to assure a fair comparison of the distinct resolutions because the number of observations and variables across the datasets of spatiotemporal models used in this study varied drastically due to different spatial and temporal resolutions. The S-Ra$^2$ statistic is a modified version of the $R^2$ statistic that adjusts for the number of predictor variables and observations in a quantitative model. It is calculated by dividing the Ra$^2$ value by the theoretical maximum value of $R^2$ for a given number of observations and predictor variables. The theoretical maximum value of $R^2$ is equal to $1 - (1/n)$, where n is the number of observations in the sample. The S-Ra$^2$ statistic ranges between 0 and 1, with higher values indicating a better fit of the model to the data after adjusting for the number of predictor variables and observations [46,47,48].

The qualitative models were evaluated based on the area under the receiver operating characteristic curve (AUC), sensitivity, and specificity. Sensitivity measures the proportion of actual positive cases that the model correctly identifies as positive. Specificity measures the proportion of actual negative cases that the model correctly identifies as negative. High sensitivity and specificity values indicate that villages with and without patients with dengue can be correctly predicted, with fewer false-negative and false-positive results, respectively.

In addition, the performance of each algorithm was evaluated by considering the computation time. All models were generated in R, version 4.1.0, with R Studio Environment, version 1.4.1717 [49]. More details regarding the characteristics of each algorithm and the model tuning parameters are presented in S1 Text and S1 Code. In the training process we repeated several times the computation with model tuning and selected the model that showed highest

accuracy. All computations were performed on a desktop computer equipped with an Intel Core i7 processor with 16 GB of random-access memory and the Windows operating system.

## Hybrid approach

We developed a hybrid approach to address the inaccuracy of quantitative predictions of dengue incidence on the weekly-village resolution (see Table 1 and Results). This approach comprises two steps, combining qualitative and quantitative models sequentially. In the first step, we transformed the dengue incidence data into a binary variable, distinguishing between zero (indicating the absence of dengue cases) and one (indicating the presence of dengue cases) values. The dataset was then divided into training (2009–2012) and test (2013) sets. Using the six algorithms, we trained a qualitative model on the training set to predict the presence or absence of dengue cases. In the second step, we fitted the quantitative model and predicted dengue incidence in all locations where the qualitative model indicated the presence of cases. Using the six algorithms, we trained the quantitative models and tested them. Following this hybrid approach, the qualitative model served as a tool for identifying villages where dengue was likely to occur, and the quantitative model was used to estimate the magnitude of cases in villages where dengue was expected to occur (Fig 1 and S1 Code).

The accuracy of the hybrid model was determined using the weighted average of the AUC and S-Ra$^2$. We used a weighted average approach to account for the varying sizes of the data used in each step of model development. The weights assigned to each measure were based on the proportion of data used to generate the respective models. Consequently, AUC was assigned a weight of 0.7 (since 70% of the data was zero-inflated), whereas S-Ra$^2$ was assigned a weight of 0.3 (since 30% of the data was non-zero inflated). This weighting scheme ensured a fair representation of the contributions from each measure, allowing for a comprehensive evaluation of the performance of the hybrid model. Because we are introducing a novel weighted average measure that combines the AUC and S-Ra$^2$ measures, we considered naming it the "Hybrid Accuracy Index" or "HAI." This name signifies the combination of both aspects of the hybrid approach and represents a unique performance measure specifically designed for this study. The code used to implement the hybrid approach can be found in S1 Code.

## Results

### Forecast of dengue incidence at different spatiotemporal resolutions

Table 1 shows the accuracy measures of the quantitative models with six spatiotemporal resolutions. Among the six ML algorithms under study, CIF yielded the best accuracy at the week-village (S-Ra$^2$ = 0.116; RMSE = 0.161), week-city (S-Ra$^2$ = 0.445; RMSE = 0.051), and month-city (S-Ra$^2$ = 0.496; RMSE = 0.040) resolutions. RF yielded the best accuracy at the month-village resolution (S-Ra$^2$ = 0.264; RMSE = 0.103). The models at the week-region and month-region resolutions predicted temporal variations only in dengue incidence across Metropolitan Manila. The highest accuracy among these resolutions was found for XGB (S-Ra$^2$ = 0.880; RMSE = 0.029) and ANN (S-Ra$^2$ = 0.718; RMSE = 0.028). Although RF exhibited a lower capacity to elucidate the variance in dengue within the week-region model, it attained the lowest error at this resolution (RMSE = 0.019). In terms of computation time, GAM was notably the fastest algorithm. All GAM models were fit in a few seconds at all resolutions for both the quantitative and qualitative models. The average computation time for the other algorithms was 0.32 h for RF, 0.73 h for XGB, 2.32 h for ANN, 2.82 h for SVM/SVR, and 3.08 h for CIF. In the quantitative models, CIF was the slowest algorithm in the week-village, week-city, and month-city datasets, computing in 6.482 h, 2.938 h, and 0.237 h, respectively. In the qualitative models, ANN was the slowest algorithm, computing in 8.514 h (Tables 1 and 2 and Fig 2).

**Table 1. Performances of the quantitative predictive models under six combinations of spatiotemporal resolutions.** The training and test observations were the data used for model training and test, respectively. The values in parentheses are the total number of zero values in each set. Max. $R^2$ is the theoretical maximum value of $R^2$ for a given number of observations and predictor variables. The computation time is the time required by the algorithm to fit the model and make predictions. The names of variables used in each model are listed in S3 Table.

| Resolution | Training Obs. | Test Obs. | Predictors | Algorithm | RMSE | $R^2$ | $Ra^2$ | Max. $R^2$ | $S\text{-}Ra^2$ | Computation time (min) |
|---|---|---|---|---|---|---|---|---|---|---|
| Week-village | 90,896 (64,917 zeros) | 22,724 (14,957 zeros) | 30 | GAM | 0.165 | 0.082 | 0.081 | 1.000 | 0.081 | 0.14 |
| | | | | RF | 0.163 | 0.103 | 0.102 | | 0.102 | 40.48 |
| | | | | CIF | **0.161** | **0.117** | **0.116** | | **0.116** | 388.92 |
| | | | | SVR | 0.179 | 0.037 | 0.036 | | 0.036 | 151.26 |
| | | | | ANN | 0.165 | 0.078 | 0.077 | | 0.077 | 384.30 |
| | | | | XGB | 0.228 | 0.104 | 0.103 | | 0.103 | 108.24 |
| Week-city | 3,536 (469 zeros) | 884 (82 zeros) | 30 | GAM | 0.052 | 0.454 | 0.435 | 1.000 | 0.435 | 0.06 |
| | | | | RF | 0.054 | 0.412 | 0.391 | | 0.391 | 1.00 |
| | | | | CIF | **0.051** | **0.464** | **0.445** | | **0.445** | 176.28 |
| | | | | SVR | 0.054 | 0.424 | 0.403 | | 0.403 | 46.82 |
| | | | | ANN | 0.052 | 0.434 | 0.414 | | 0.414 | 10.62 |
| | | | | XGB | 0.173 | 0.379 | 0.357 | | 0.357 | 4.89 |
| Week-region | 208 (1 zero) | 52 (No zeros) | 8 | GAM | 0.026 | 0.888 | 0.867 | 0.996 | 0.870 | 0.01 |
| | | | | RF | **0.019** | 0.884 | 0.863 | | 0.866 | 1.16 |
| | | | | CIF | 0.021 | 0.853 | 0.826 | | 0.829 | 1.69 |
| | | | | SVR | 0.026 | 0.817 | 0.783 | | 0.786 | 0.13 |
| | | | | ANN | 0.027 | 0.844 | 0.815 | | 0.818 | 0.14 |
| | | | | XGB | 0.029 | **0.896** | **0.877** | | **0.880** | 1.88 |
| Month-village | 20,976 (9,474 zeros) | 5,244 (1,899 zeros) | 30 | GAM | 0.106 | 0.179 | 0.174 | 1.000 | 0.174 | 0.20 |
| | | | | RF | **0.103** | **0.268** | **0.264** | | **0.264** | 22.68 |
| | | | | CIF | **0.103** | 0.262 | 0.258 | | 0.258 | 408.12 |
| | | | | SVR | 0.110 | 0.187 | 0.183 | | 0.183 | 528.78 |
| | | | | ANN | 0.107 | 0.173 | 0.169 | | 0.169 | 66.06 |
| | | | | XGB | 0.186 | 0.207 | 0.202 | | 0.202 | 39.24 |
| Month-city | 816 (18 zeros) | 204 (6 zeros) | 30 | GAM | 0.046 | 0.524 | 0.442 | 0.999 | 0.442 | 0.01 |
| | | | | RF | 0.041 | 0.539 | 0.459 | | 0.459 | 0.00 |
| | | | | CIF | **0.040** | **0.570** | **0.496** | | **0.496** | 14.22 |
| | | | | SVR | 0.045 | 0.486 | 0.396 | | 0.396 | 2.77 |
| | | | | ANN | 0.046 | 0.481 | 0.391 | | 0.391 | 2.59 |
| | | | | XGB | 0.169 | 0.412 | 0.310 | | 0.310 | 1.95 |
| Month-region | 48 (No zeros) | 12 (No zeros) | 8 | GAM | 0.028 | 0.910 | 0.669 | 0.983 | 0.680 | 0.00 |
| | | | | RF | **0.017** | 0.916 | 0.692 | | **0.704** | 0.08 |
| | | | | CIF | 0.026 | 0.855 | 0.669 | | 0.680 | 0.04 |
| | | | | SVR | 0.023 | 0.857 | 0.477 | | 0.485 | 0.06 |
| | | | | ANN | 0.028 | **0.920** | **0.706** | | 0.718 | 0.63 |
| | | | | XGB | 0.028 | 0.890 | 0.596 | | 0.606 | 2.01 |

## Qualitative forecast of the absence or presence of dengue

The qualitative predictive model was developed using the week-village resolution data. RF yields the best overall accuracy (AUC = 0.799). Meanwhile, SVM showed better sensitivity (0.779), while XGB showed better specificity (0.751) (Table 2).

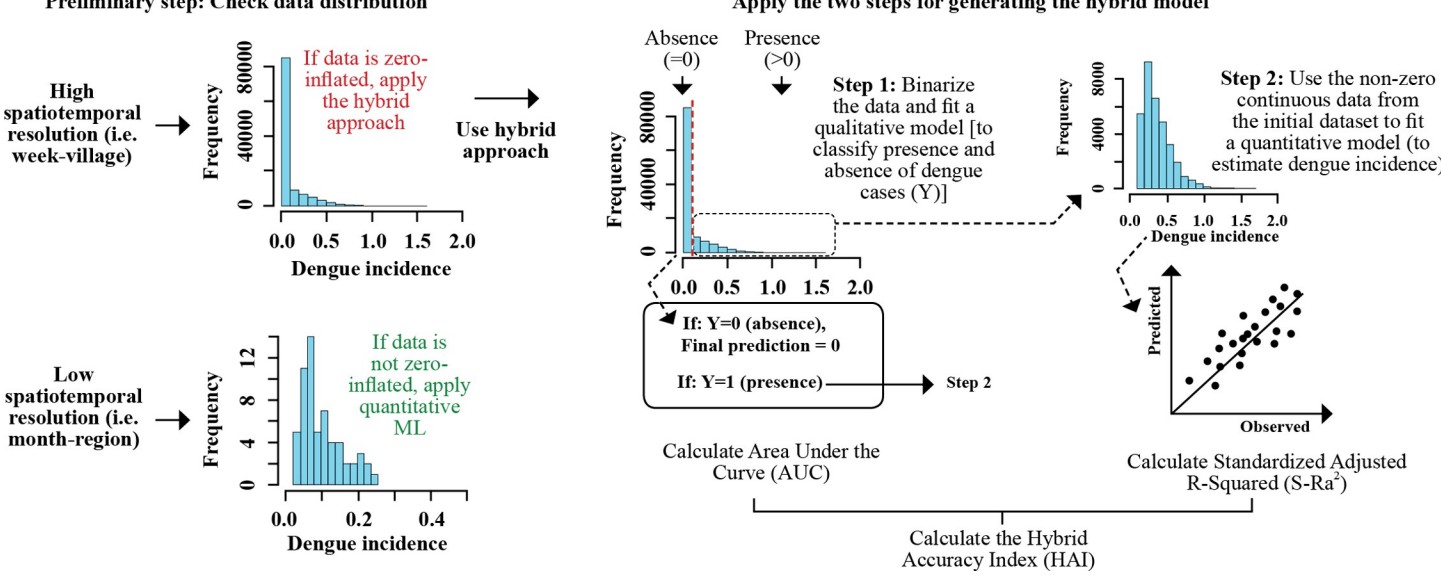

**Fig 1. Conceptual framework for the implementation of the hybrid model.**

## Village-scale temporal models

For the quantitative models, 10%, 36%, and 54% of the villages showed $R^2$ values ranging from 0.3 to 0.8, from 0.1 to 0.3, and from 0 to 0.1, respectively (Fig 3A). For the qualitative models, the villages were classified into three groups: high AUC (0.75–1.0), moderate AUC (0.5–0.75), and low (AUC < = 0.5). The proportion of villages per group was 32%, 64%, and 4%, respectively (Fig 3C).

The variables that showed significant Spearman's correlation coefficient with model $R^2$ and AUC were dengue cases, dengue incidence, the frequency of dengue cases, village area, village population, the proportion of low residential density, and open spaces (Table 3).

## Improvement of accuracy with merging small villages

Merging small villages in the Week-village data to increase the variance in predictor values (i.e. precipitation) improved the predictive accuracy for quantitative models with all algorithms but not in the qualitative models. For example, CIF improved from S-Ra$^2$ = 0.116 to 0.220 for the quantitative model and from AUC = 0.794 to 0.806 in the qualitative model with merged villages. Similar improvements were also observed with GAM and ANN (S4 and S5 Tables).

**Table 2. Performance of the qualitative models with the week-village dataset.** Training and Test Observations are observations were the subsets used for model training and test, respectively. The values in parentheses are the total number of zero values in each set. The computation time is the time taken by the algorithm to complete model estimation and prediction. The names of variables used in each model are listed in S3 Table.

| Training Obs. (64,917 zeros) | Test Obs. (14,957 zeros) | Predictors | Algorithm | AUC | Sensitivity | Specificity | Computation time (min) |
|---|---|---|---|---|---|---|---|
| 90,896 | 22,724 | 30 | GAM | 0.689 | 0.591 | 0.675 | 0.57 |
| | | | RF | **0.799** | 0.748 | 0.745 | 68.88 |
| | | | CIF | 0.794 | 0.732 | 0.745 | 306.12 |
| | | | SVM | 0.753 | **0.779** | 0.717 | 454.56 |
| | | | ANN | 0.715 | 0.699 | 0.692 | 510.84 |
| | | | XGB | 0.791 | 0.717 | **0.751** | 146.70 |

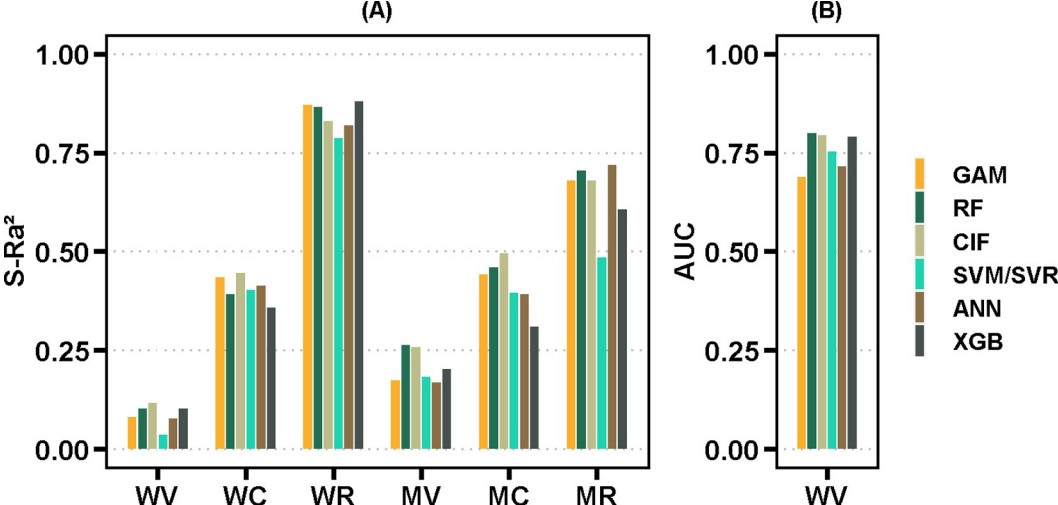

**Fig 2. Model accuracy comparison summary.** (A) Are the quantitative models with the six spatiotemporal resolutions and (B) are the qualitative models with the week-village resolution. WV is week-village, WC is week-city, WR is week-region, MV is month-village, MC is month-city and MR is month-region.

However, with qualitative models, RF, SVM and XGB showed reduced accuracy with aggregated data. The data with aggregated villages was also used for generating individual village temporal models. With the quantitative model, 28% of villages showed an accuracy ($R^2$) between 0.3–0.8, 42% from 0.1–0.3 and 30% from 0.0–0.1 (Fig 3A). With qualitative model, 35% of villages achieved accuracy (AUC) between 0.75–1.0, 63% from 0.5–0.75, and 2% with AUC $< = 0.5$ (S3 Fig).

## Improvement in accuracy with the hybrid model

Table 4 presents the accuracy measures for the hybrid model. CIF yielded an accuracy of 0.771, followed by RF with 0.559, XGB with 0.591, SVM with 0.575, ANN with 0.510, and GAM with 0.491. Although the quantitative model could only explain a maximum of 11.6% of the variations in dengue incidence (Table 1), the second step of the hybrid model could explain approximately 71.6% (Table 4) of the variations in dengue incidence in villages expected to be dengue positive from the first step. For the hybrid model, in the first step, we used 90,896 observations for training and 22,724 observations for testing the qualitative models. In the second step for the quantitative models, the number of observations for training was 25,979 and that for testing varied depending on the prediction by each algorithm in the first step. Therefore, GAM was tested with 1,419 observations, RF with 4,012 observations, CIF with 4,145 observations, SVM with 2,674 observations, ANN with 1,954 observations, and XGB with 4,482 observations. All models were fit with the same number of predictors, which was 30.

## Discussion

This study introduces a hybrid approach that combines both binary and continuous outcome modeling to enhance predictive accuracy in dengue forecasting. This method builds upon and complements existing methodologies such as zero-inflated and hurdle models. Our initial findings revealed that models with coarser spatial and temporal resolutions tended to exhibit better accuracy than those with finer resolutions. This suggests that the quality of data, such as low variance and non-normally distributed dengue incidence, may influence the accuracy of ML in dengue forecasting using RS data at a finer resolution. The hybrid approach

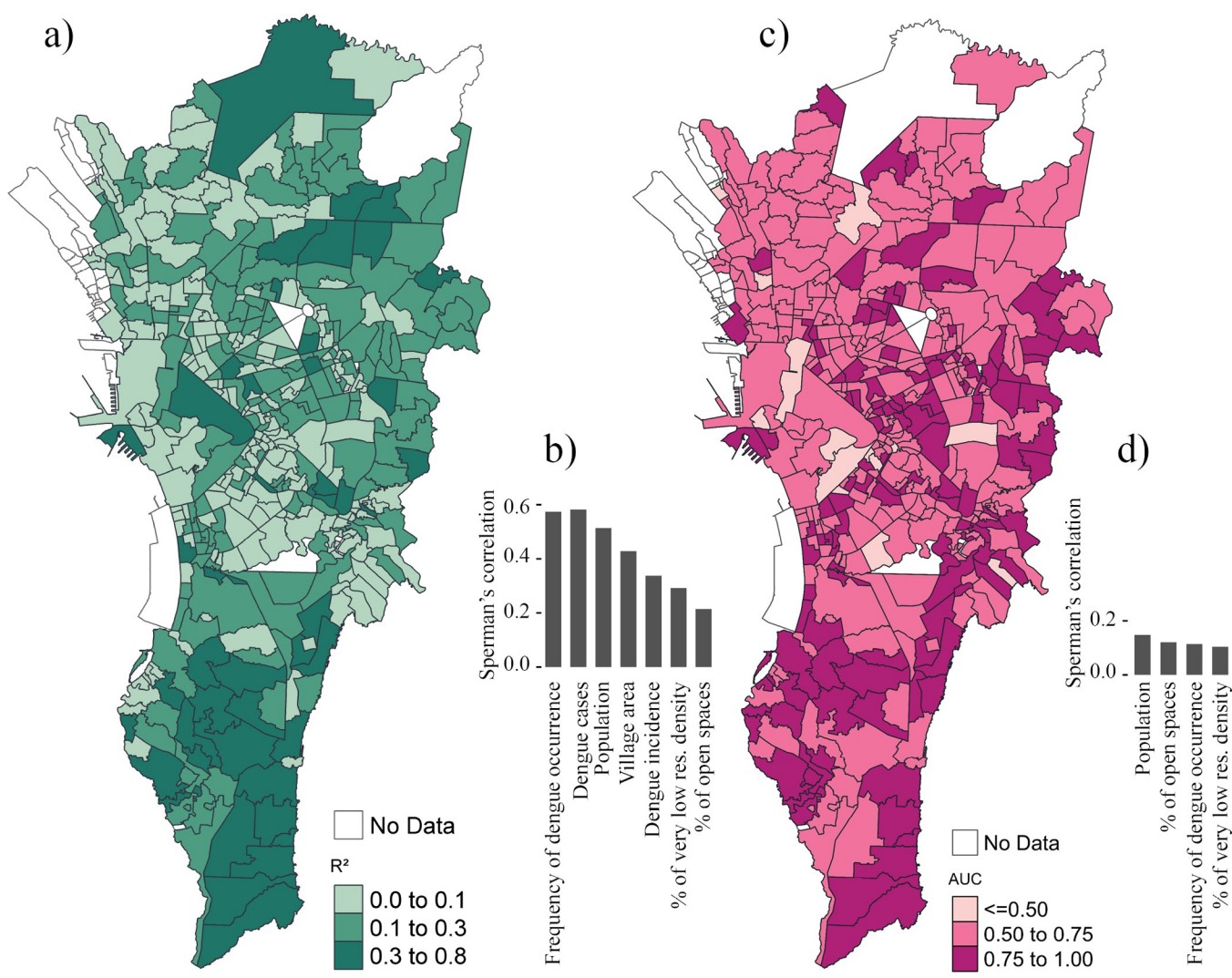

**Fig 3. Accuracy of the temporal models at each village.** (a) quantitative and (c) qualitative models. Variables that demonstrated a significance in Spearman's correlation statistical test ($p < 0.05$) between (b) $R^2$ and (d) AUC to dengue cases, dengue incidence, frequency of dengue cases, village area, population, and proportion of land-use types. The names of variables used in each model are listed in S3 Table. The maps' baselayer are village boundaries obtained from https://data.humdata.org/dataset/cod-ab-phl. The license terms can be found here https://data.humdata.org/faqs/terms. The maps were created using the "*tmap*" package in R [49].

demonstrated promising results in improving model accuracy at finer spatiotemporal resolution with continuous data, where data may be inflated with zero values. Because six ML algorithms were used in this study, the following discussion regarding the comparison of model accuracy will focus solely on the best accuracy score per resolution dataset, regardless of the ML algorithm used.

Our study utilized climatic and landscape data, which are well-established and effective predictors of dengue incidence. Climatic factors significantly impact the transmission of dengue by influencing the life cycle and behavior of mosquitoes. For instance, higher temperatures accelerate the mosquito life cycle and virus replication, reducing the extrinsic incubation period and increasing the risk of outbreaks. Similarly, environments with high relative humidity, especially above 60%, promote mosquito survival and activity, thereby increasing their contact rate with humans. Rainfall creates temporary water bodies ideal for mosquito

**Table 3. Spearman's correlation analysis between quantitative and qualitative model accuracy ($R^2$ and AUC) with dengue cases, dengue incidence, frequency of dengue cases, village area, village population, and proportion of land-use types.** The *p*-values for the quantitative model should not be considered exact values because some values were tied in the data used for computing Spearman's rank correlation coefficient. (-) Values not shown because correlation was insignificant.

| Variable | $R^2$ of quantitative models | | AUC of the qualitative models | |
|---|---|---|---|---|
| | $r_s$ | *P*-value | $r_s$ | *P*-value |
| Frequency of dengue cases | 0.57 | <0.001 | 0.12 | 0.017 |
| Temporal mean dengue cases | 0.57 | <0.001 | - | - |
| Population size | 0.51 | <0.001 | 0.14 | 0.006 |
| Village area | 0.43 | <0.001 | - | - |
| Temporal mean dengue incidence | 0.34 | <0.001 | - | - |
| % of very low residential density | 0.28 | <0.001 | 0.11 | 0.020 |
| % open spaces | 0.21 | <0.001 | 0.12 | 0.014 |

breeding, but excessive rainfall can wash away larvae, reducing populations temporarily. Conversely, a lack of rainfall often results in water being stored in containers, which can become mosquito breeding grounds if not properly managed. In residential areas, high human density, along with water storage and discarded items, enhances mosquito-human contact and breeding opportunities. Furthermore, improper waste disposal and water storage in commercial and industrial areas can also lead to breeding sites, with the movement of people between these and residential areas facilitating the virus's spread. Even though less directly linked, vegetated areas like parks and forests can still harbor mosquito populations, particularly if they contain water features that support mosquito breeding [26]. Although our data incorporated significant predictive factors for dengue incidence, achieving high accuracy at fine spatial scales proved challenging due to other factors discussed below.

## Data quality interference in the prediction of dengue incidence using quantitative models

Low model accuracy at the week-village resolution seemed to be influenced by the characteristics of our data, which have a high frequency of zero values in dengue incidence (64,917 zeros of 90,896 observations in the training data), generating the so-called zero-inflated data [50]. The accuracy of the quantitative models improved when the spatial resolution was reduced from village to city or from city to region. The same trend was observed when the temporal resolution was reduced from week to month. Notably, the increase in accuracy was more pronounced with spatial resolution reduction compared to temporal resolution reduction. This phenomenon could be attributed to the greater efficacy of data aggregation in the spatial dimension in reducing zero values compared to aggregating in the temporal dimension (S1 Fig).

**Table 4. Accuracy of the hybrid model.** HAI is the Hybrid Accuracy Index. The names of variables used in each model are listed in S3 Table.

| Model | AUC of the qualitative models | S-Ra$^2$ of quantitative models | HAI |
|---|---|---|---|
| GAM | 0.689 | 0.022 | 0.491 |
| RF | 0.799 | 0.127 | 0.599 |
| CIF | 0.794 | 0.716 | **0.771** |
| SVM | 0.753 | 0.155 | 0.575 |
| ANN | 0.715 | 0.024 | 0.510 |
| XGB | 0.791 | 0.119 | 0.591 |

In ML, models trained on datasets with high-frequency values (e.g., zero in dengue incidence in this study) tend to exhibit high accuracy in predicting high-frequency values. However, this ML property holds true only if there is a high correlation between the dependent variable and its predictors [51,52]. The correlation between dengue incidence and environmental factors appeared to be low in the week-village resolution. This is because the incidence of dengue remained at the same level (i.e., 0) for several weeks despite changes in environmental conditions (e.g., temperature, precipitation, vegetation, and wind speed). This made it challenging for ML models to generate models with high accuracy and may have led to inaccurate predictions even for the majority values (e.g., zero dengue incidence). Notably, all six ML algorithms could accurately predict moderate levels of dengue incidence at the week-village resolution but failed to accurately predict extreme values (e.g., zero). Although ML algorithms can deliver optimal prediction accuracy in large data even in the presence of multicollinearities and outliers [1], their performance in continuous data with zero-inflated properties remains unclear. Moreover, the black-box nature of ML algorithms limits the comprehension of their behavior with zero-inflated data [19]. In epidemiological data on infectious diseases, especially at fine spatial and temporal resolutions, the frequency of zero values is typically high, except during a pandemic or when the disease becomes epidemic. Therefore, alternative approaches for improving prediction accuracy in zero-inflated data with ML remain to be investigated.

The low model accuracy in the week-village resolution can also be attributed to the low variances in both dengue incidence and environmental factors. The variance of data plays a critical role in the accuracy of ML algorithms because it determines the extent to which the independent variables affect the dependent variable [53]. If the dependent variable has sufficient variance, the model can be trained from more examples of different possible outcomes for a given set of inputs. This allows the model to make predictions that are more likely to generalize to the new data by achieving an optimal bias–variance tradeoff [54]. However, the variance of dengue incidence was low, with the observations of the dependent variable clustered closely around the same value (i.e., zero). Because some RS data, such as precipitation, wind speed, and RH, were only available at a spatial resolution of about 10 km (pixel area of 100 $km^2$), variance in the predictor variables was also limited. Over 96% of the villages (420 of 437 villages) had an area of <5 $km^2$, requiring an assumption that all small villages within a 100-$km^2$ pixel would have consistent values of RS data and null variance. The lower spatial environmental heterogeneity of the RS predictors may explain the failure of the models to obtain high accuracy in the week-village resolution. Our approach for approximating the spatial resolution of the administrative units to the pixel size by merging small villages (S4 and S5 Tables) or by using city and regional scales increased the variance of predictors, which may have improved the ability of the algorithms to fit quantitative models [55]. This suggests that the scale ratio of administrative units and the pixel size of RS data is significantly important in spatial modeling [56] and may be a factor that limits model accuracy for spatial prediction modeling.

Furthermore, dengue cases were reported from passive surveillance through healthcare facilities. The number of cases may be lower than the actual numbers and may not be sufficient to create accurate predictive models. Passive surveillance cannot determine all cases and may introduce bias in the data [57,58]. Furthermore, data may be biased toward individuals who seek medical care when sick [59]. On the other hand, the proportion of symptomatic cases seeking medical care could also vary spatially and temporally base on access to health care This can lead to incomplete or biased data that may not represent the broader population. As per a previous study, the proportion of unnoticed individuals without symptoms carrying the dengue virus could reach 50%–90% of all infection cases in a population [60]. These infections can be detected only if active serological surveillance is employed [61]. Since the data commonly

used for implementing the predictive models in epidemiology including in this study is the number of cases of those seeking medical care, continuous enhancements in modeling approaches are crucial for refining predictions, given the inherent constraint that not all infections exhibit severe symptoms. Addressing these challenges and achieving accurate predictions, particularly for the number of infections requiring medical care, is vital for healthcare preparedness. This may entail exploring additional data types, such as herd immunity from previous infections or the heterogeneity in the degree of exposure within the populations, to reinforce the effectiveness of predictive methodologies.

## Comparison of the quantitative and qualitative models

Although 79.9% of the spatiotemporal variations in dengue occurrence in the week-village resolution were explained by the qualitative model, the percentage of variations in dengue incidence explained by the quantitative model was only 11.6%. This difference seems to be specific to this study because of the skewed distribution of dengue incidence data, which negatively affected the performance of quantitative models. Despite the data limitations, the qualitative models displayed better accuracy than the quantitative models. This can be attributed, in part, to the simplicity of qualitative models, which only have a finite number of possible values for the dependent variable because of a narrower decision boundary with only two possible outcomes. In contrast, quantitative models involve predicting a continuous variable that can take an infinite number of values. The optimal model parameters in qualitative models can be estimated by optimizing a suitable loss function, such as cross-entropy, for separating two classes only [62]. In contrast, in quantitative models, the goal is to learn a function that maps input variables to a random number output, which often requires solving a parameter optimization problem that involves minimizing a loss function, such as mean squared error [63]. This optimization problem can be more computationally intensive, particularly when dealing with zero-inflated data, where non-zero dengue incidence observations are very rare [64].

## Characteristics of villages in which temporal predictions tend to be inaccurate

The prediction accuracy of the temporal dynamics of dengue incidence in each village was positively correlated with population size, village area, number of dengue cases, and frequency of dengue cases in both quantitative and qualitative models. These results may reflect the same phenomenon where villages with a large area tend to have a larger population. We considered that the population size per village may be an important factor in determining the accuracy of the model. In villages with small populations, when dengue incidence is low, it is more likely that few dengue cases would be missed (i.e., counted as zero cases) during passive surveillance. The high frequency of false-negative data occurring in low dengue incidence makes it difficult to correctly extract the true temporal variation patterns using ML. In contrast, villages with large populations are more likely to capture dengue cases even when dengue incidence is low and thus may more accurately extract temporal patterns.

## The two-step hybrid approach for tackling zero-inflated data

The hybrid approach used in this study effectively addressed the challenges of predicting zero-inflated data. In the hybrid approach, if the first-step qualitative model predicts one (i.e., dengue case present), the magnitude of dengue incidence is estimated by the second-step quantitative model. Filtering out all zero predictions in the first step improved the accuracy of the quantitative model. In contrast, the qualitative model played a crucial role in identifying villages and weeks in which dengue is likely to be absent. By accurately identifying villages and

weeks in which dengue is likely to occur, the qualitative model played a crucial role in guiding targeted interventions. This approach allowed for a more efficient orientation of efforts and interventions by directing them toward regions with a higher probability of dengue occurrence. By focusing efforts on these high-risk areas, the model helps ensure that interventions are timely and strategically implemented where they are most needed, maximizing their effectiveness and impact.

Emphasizing the technical significance of the hybrid approach is important. By excluding zero predictions in the first step, we refined the dataset for the subsequent quantitative model, thus mitigating the challenges associated with zero-inflated data. This preprocessing step enhanced the ability of the quantitative model to accurately capture and estimate the magnitude of dengue incidence. The hybrid approach can be viewed as a complementary framework that combines the strengths of the qualitative and quantitative models. The qualitative model excels in handling data with numerous zeros but lacks quantitative capabilities, whereas the quantitative model is adept at quantification but can struggle with datasets containing many zero values. However, false negatives may arise from the first step with implications to management programs, requiring further work for addressing the false negatives. Regarding the performance measure calculation, we assigned weights based on the proportion of data used in each step. These weighted performance measures ensured a fair representation of their respective contributions in the hybrid approach, thus providing a comprehensive evaluation.

Upsampling and oversampling techniques have been proposed for improving accuracy in imbalanced data [65]. When applying upsampling or downsampling, the model accuracy can be significantly influenced by the choice of sampling sizes, which can introduce uncertainties regarding the stability of the model [66]. In our qualitative models, we observed fair model accuracy and therefore did not find it necessary to apply upsampling or downsampling. However, the framework of the hybrid approach is designed to be generalizable and flexible, allowing for the incorporation of various techniques (i.e. upsampling, downsampling, synthetic minority oversampling technique) to meet the specific modeling circumstances and characteristics of the data. This study's proposed hybrid model adheres to the same framework as the zero-inflated [20] and hurdle [21] models, utilizing a two-step forecasting process that advances to quantitative forecasting only if 'presence' is predicted in the initial qualitative forecast. What sets this study apart is its pioneering effort to integrate machine learning within this framework. Moreover, the hybrid model's implementation process enables the switching between different machine learning algorithms at each step. While both steps of our hybrid model utilized the same algorithm, it is designed with the flexibility to use different algorithms at each step–for example, one algorithm for classifying zero/non-zero observations and another algorithm for quantifying continuous outcomes in the second step.

Although our study specifically focused on dengue prediction, the novel hybrid approach has broader applicability. Beyond the field of epidemiology, this approach can be applied to various domains where predictive modeling encounters challenges associated with zero-inflated data (i.e., species occurrence, environmental pollution, accidents, and gene expression). The new weighted accuracy metric we proposed, i.e., the HAI, can combine the traditional accuracy measures used in ML for qualitative and quantitative models. By harnessing the power of ML algorithms and integrating qualitative and quantitative models, our hybrid approach can address the zero-inflated data problem and improve prediction accuracy across diverse disciplines. Furthermore, given that machine learning algorithms can handle a variety of distributions, including Negative Binomial, Poisson, and Gaussian, we believe that the second step of our hybrid approach could be applied to count data. However, it's important to note that count data was not used in this study. Future research might explore implementing the hybrid approach with count data and compare the outcomes with traditional hurdle models.

### Accuracy of ML algorithms in different datasets

CIF and RF produced more accurate and comparable predictions overall. Although RF was systematically accredited to outperform other ML algorithms [2,67,68], CIF was found to be superior to RF, particularly in datasets with a higher number of observations. The perception that RF outperforms many other ML algorithms may be attributed to the lack of comparative studies between RF and CIF. Studies directly comparing RF with CIF are scarce [69]. Although CIF and RF are ensemble methods that combine multiple decision trees to make predictions, CIF uses statistical tests to determine the optimal split at each node in a decision tree. This helps improve accuracy using only important variables. In contrast, RF uses a random variable selection method for growing trees and does not perform statistical tests, which may result in less accurate splitting decisions. The enhanced accuracy of CIF can be linked to its strategy of using only the most relevant variables [70]. Among the ML algorithms under study, CIF was the only algorithm that could perform variable selection by minimizing the $p$-value of a conditional inference independency test when fitting the model.

The increased accuracy of CIF is accompanied by high computation costs compared with RF. The algorithm training process of CIF to produce unbiased trees is time-consuming because several permutation tests are required [71]. These permutation tests involve repeating the generation of each individual tree by permuting the response to retrieve a $p$-value for assessing the strength of variables before splitting. In RF, the partitioning with variables to generate trees does not involve permutation tests. However, the interpretation and comparison of the computation time between algorithms should be performed with care. Computation time may depend on the number and dimension of parameters used to tune the model. In contrast, the hardware, software, and existing background or parallel processing tasks may differ and interfere with computation time [72]. In the case of the matrix of the parameters, the more the dimension is increased, the more the optimization of the model intensifies, thus increasing the computation time [73]. Considering that each algorithm has a different structure, it is expected that the configuration of these parameters will greatly influence the computation time. Furthermore, it may also be affected by the R package used for the model. Because each algorithm can be available from different packages (or even package wrappers), some packages will be faster than others even for the same algorithm [74].

Unfortunately, more recent data was not accessible during the study, limiting our analysis to 2009–2013 timeframe. However, because dengue occurrence in Metropolitan Manila is endemic with seasonal peaks following the rainy season (S4 Fig), we strongly believe that the performance of the hybrid model would not vary significantly with more recent data. Nonetheless, future studies with more recent data are worth conducting. On the other hand, future research should also consider comparing the proposed hybrid approach with other similar approaches such as hurdle, mixed, or zero-inflated models with datasets from different domains.

## Conclusions

This study introduces a hybrid approach that combines both binary and continuous outcome modeling to enhance predictive accuracy in dengue forecasting. In the context of Metropolitan Manila, the RS-based climatic predictors may be effective for qualitative prediction of dengue incidence up to the village scale with standard approaches. Quantitative dengue incidence prediction may be performed at the city or regional scale. We found that the increased frequency of zero dengue incidence at fine temporal and spatial resolution data may lead to inaccurate predictions. For these scenarios, the hybrid approach may significantly improve the prediction accuracy in these zero-inflated continuous data where zero-inflated or hurdle models cannot

be used. Models using CIF and RF performed better; however, other algorithms may be able to predict dengue infections depending on the characteristics of the dataset (i.e., highly correlated data). Note that the results presented in this study may reflect the situation of Metropolitan Manila because the size of administrative units may vary per location. Nonetheless, they showed critical interference of data resolution on ML model accuracy. These results are essential for developing dengue surveillance, prevention, and control strategies and highlight important considerations for future research on predicting with zero-inflated data.

## Supporting information

**S1 Fig. Distribution of Dengue Incidence across different spatiotemporal resolutions.** DI means Dengue Incidence.
(TIF)

**S2 Fig. Flow chart for data acquisition and processing.**
(TIF)

**S3 Fig. Temporal models accuracy changes with aggregating villages.** The maps baselayer are village boundaries obtained from https://data.humdata.org/dataset/cod-ab-phl. The license terms can be found here https://data.humdata.org/faqs/terms. The maps were created using the "*tmap*" package in R.
(TIF)

**S4 Fig. Spatiotemporal Dengue Incidence Heatmaps.**
(TIF)

**S1 Table. Characteristics of satellite data obtained from Google Earth Engine.**
(DOCX)

**S2 Table. Weekly and monthly cross-correlation analysis of predictors to dengue incidence.**
(DOCX)

**S3 Table. Variables used for the implementation of models.**
(DOCX)

**S4 Table. Quantitative model accuracy changes with aggregating villages.**
(DOCX)

**S5 Table. Qualitative model accuracy changes with aggregating villages.**
(DOCX)

**S1 Text. Model approach and implementation.**
(DOCX)

**S1 Code. Code for the implementation of the hybrid model.**
(DOCX)

**S1 Video. Reconstructed weekly MODIS-Terra daytime LST.** The maps baselayer are village boundaries obtained from https://data.humdata.org/dataset/cod-ab-phl. The license terms can be found here https://data.humdata.org/faqs/terms. The maps were created using the "*tmap*" package in R. The animation was created in GRASS GIS.
(MP4)

## Acknowledgments

We would like to acknowledge Katherine M. Viacrusis, Lara Fides T. Hernandez, and Howell T. Ho from Trinity University of Asia for collecting and providing the data on notified dengue fever cases.

## Author Contributions

**Conceptualization:** Kozo Watanabe.

**Data curation:** Micanaldo Ernesto Francisco, Thaddeus M. Carvajal.

**Formal analysis:** Micanaldo Ernesto Francisco.

**Funding acquisition:** Kozo Watanabe.

**Investigation:** Micanaldo Ernesto Francisco.

**Methodology:** Micanaldo Ernesto Francisco, Kozo Watanabe.

**Project administration:** Kozo Watanabe.

**Resources:** Kozo Watanabe.

**Software:** Micanaldo Ernesto Francisco.

**Supervision:** Kozo Watanabe.

**Validation:** Micanaldo Ernesto Francisco, Kozo Watanabe.

**Visualization:** Micanaldo Ernesto Francisco.

**Writing – original draft:** Micanaldo Ernesto Francisco.

**Writing – review & editing:** Micanaldo Ernesto Francisco, Thaddeus M. Carvajal, Kozo Watanabe.

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
