## [Decision Letter · Decision Letter 0]

12 Feb 2024

Dear Dr. Watanabe,

Thank you very much for submitting your manuscript "Hybrid Machine Learning Approach to Zero-Inflated Data Improves Accuracy of Dengue Prediction" for consideration at PLOS Neglected Tropical Diseases. As with all papers reviewed by the journal, your manuscript was reviewed by members of the editorial board and by several independent reviewers. In light of the reviews (below this email and in the attached document), we would like to invite the resubmission of a significantly-revised version that takes into account the reviewers' comments. 

We cannot make any decision about publication until we have seen the revised manuscript and your response to the reviewers' comments. Your revised manuscript is also likely to be sent to reviewers for further evaluation.

Sincerely,

Christopher M. Barker

Academic Editor

Victoria Brookes

Section Editor

Reviewer's Responses to Questions

**Key Review Criteria Required for Acceptance?**

**Methods**

-Are the objectives of the study clearly articulated with a clear testable hypothesis stated?

-Is the study design appropriate to address the stated objectives?

-Is the population clearly described and appropriate for the hypothesis being tested?

-Is the sample size sufficient to ensure adequate power to address the hypothesis being tested?

-Were correct statistical analysis used to support conclusions?

-Are there concerns about ethical or regulatory requirements being met?

Reviewer #1: Some sections of the Methods would benefit from further details and clarifications.

Line 150. Why did you choose 2009-2013 for the model? Was more recent data not available? Please provide a little more detail on the rationale for the chosen time period.

Lines 237-245. The description of these temporal models is confusing. How do they differ from the previously described six temporal-spatial combination models? Also, which ML algorithm(s) were used for these models?

Line 275. What was your rational for using a different train/test method for the hybrid model (80/20 split of 2009-2013 data) vs. the qualitative and quantitative models (2009-2012 train and 2013 test)? Comparison of model performance resulting from different test/train methodologies seems potentially problematic.

Line 277. The way you’ve appeared set up the hybrid model means that it could not be used in a predictive manner. The description seems to indicate that the quantitative models in the second step is only applied to locations where nonzero case counts occurred (not locations predicted to have cases from step 1). This seems like a big limitation to the future use of the proposed hybrid model. Fitting a quantitative model in all locations where the qualitative model indicated the presences of cases could overcome this limitation. Based on Line 360, this may actually have been what was done. Please clarify.

Line 290. Did you consider evaluating the hybrid models in a similar manner as other hurdle models? Absence locations can be treated as zeros so the full output from both steps can be evaluated as a quantitative model (e.g., calculate RMSE, R2-values). This could be an additional metric to compare to your proposed HAI metric.

Reviewer #2: (No Response)

Reviewer #3: The authors present the methods clearly, the methods are well thought out and include a defensible and reasonable combination of models and tests for accuracy. The statistical analysis is robust. It might be nice to have slightly more description of the "quantitative" model, where all the zeros are taken out. Does this allow for the quantitative model using previous dengue case counts as one potential predictor for future cases?

**Results**

-Does the analysis presented match the analysis plan?

-Are the results clearly and completely presented?

-Are the figures (Tables, Images) of sufficient quality for clarity?

Reviewer #1: Consider developing an additional figure which would visually present the model comparison results from the Tables in the main text. This would aid the reader in comparing accuracy across modeling techniques and spatio-temporal scales, a major takeaway of your manuscript.

Reviewer #2: (No Response)

Reviewer #3: The results are clearly presented and match the analysis as described.

**Conclusions**

-Are the conclusions supported by the data presented?

-Are the limitations of analysis clearly described?

-Do the authors discuss how these data can be helpful to advance our understanding of the topic under study?

-Is public health relevance addressed?

Reviewer #1: Conclusions are supported by the presented results. Contextualization of the proposed hybrid model in terms of currently used hurdle, mixture, and zero-inflated modeling frameworks would improve the discussion.

Reviewer #2: (No Response)

Reviewer #3: Yes, the conclusions are clear and described well. The authors provide a creative and really useful hybrid approach that clearly improves predictions for dengue, particularly at smaller spatial/temporal scales.

**Editorial and Data Presentation Modifications?**

Reviewer #1: Line 172 (and others). Your indication of P-values could be misinterpreted as being truly zero. It would be more appropriate to indicate these values as something like <0.001. This would also apply to the Tables.

Line 208. You state “weeks” twice in this sentence. Likely you meant “month” for the seven time points.

Reviewer #2: (No Response)

Reviewer #3: A few minor typos in references, and I don't think you need to describe the words "spatial" and "temporal" in the first paragraph (i.e. take out the parentheses)

**Summary and General Comments**

Reviewer #1: Francisco et al. applied six machine learning algorithms to predict dengue cases in Metropolitan Manila in 2013. Their comparison of performance across spatio-temporal scales and between algorithms are highly valuable contributions. The authors also described a hybrid modeling approach to first predict the presence/absence of cases and then the number of dengue cases in “present” locations, finding an increase in accuracy with this method. While this framework may be new when applied to machine learning models, this is not a conceptually new framework; hurdle, mixture, and zero-inflated techniques are currently employed across a range of settings. Providing some discussion and comparison of the proposed hybrid model with such frameworks would frame the discussion.

Reviewer #2: (No Response)

Reviewer #3: In general, this is a useful, creative approach to examining the spatial-temporal scales at which different models are more accurate and using a new hybrid approach that the authors developed to leverage the accuracy of the "qualitative" 0-1 models with the more useful case counts predictions of quantitative models that don't do well with a lot of zeros. Very cool!!

PLOS authors have the option to publish the peer review history of their article (what does this mean?). If published, this will include your full peer review and any attached files.

Reviewer #1: No

Reviewer #2: No

Reviewer #3: No
---

## [Decision Letter · Decision Letter 1]

1 Aug 2024

Dear Dr. Watanabe,

Thank you very much for submitting your manuscript "Hybrid Machine Learning Approach to Zero-Inflated Data Improves Accuracy of Dengue Prediction" for consideration at PLOS Neglected Tropical Diseases. As with all papers reviewed by the journal, your manuscript was reviewed by members of the editorial board and by several independent reviewers. The reviewers appreciated the attention to an important topic. Based on the reviews, we are likely to accept this manuscript for publication, providing that you modify the manuscript according to the review recommendations. 

Please consider moderating claims about the high degree of novelty or pioneering nature of this paper, as some seem exaggerated. Incremental progress has value, and we agree that this study is a useful advance. Modeling and prediction of dengue at various spatio-temporal scales with a wide range of model forms, including zero-inflated and machine learning models, have been published previously. It is somewhat misleading to claim that "this is the first instance in which dengue prediction accuracy has been compared across various spatiotemporal resolutions." Even in the unlikely event that this is true, collectively many other studies on dengue prediction have evaluated predictive performance across a very wide range of spatio-temporal resolutions.

Sincerely,

Christopher M. Barker

Academic Editor

Victoria Brookes

Section Editor

From the editors:

Please consider moderating claims about the high degree of novelty or pioneering nature of this paper, as some seem exaggerated. Incremental progress has value, and we agree that this study is a useful advance. Modeling and prediction of dengue at various spatio-temporal scales with a wide range of model forms, including zero-inflated and machine learning models, have been published previously. It is somewhat misleading to claim that "this is the first instance in which dengue prediction accuracy has been compared across various spatiotemporal resolutions." Even in the unlikely event that this is true, collectively many other studies on dengue prediction have evaluated predictive performance across a very wide range of spatio-temporal resolutions.

Reviewer's Responses to Questions

**Key Review Criteria Required for Acceptance?**

**Methods**

-Are the objectives of the study clearly articulated with a clear testable hypothesis stated?

-Is the study design appropriate to address the stated objectives?

-Is the population clearly described and appropriate for the hypothesis being tested?

-Is the sample size sufficient to ensure adequate power to address the hypothesis being tested?

-Were correct statistical analysis used to support conclusions?

-Are there concerns about ethical or regulatory requirements being met?

Reviewer #1: Line 240. Using the whole timeseries in the cross-correlation analysis could have inadvertently “leaked” information from the training to the testing set, thus artificially improving the testing performance of the models. Why was the whole timeseries used and how did you check you had not introduced this bias?

Reviewer #2: My primary remaining question with regards to the approach used here deals with one of the arguments that the authors present as a primary reason for using their two-stage ML approach instead of a more "traditional" (i.e. parametric) hurdle model. Namely, in L125 the authors argue that zero-inflated and hurdle models are designed for count data, but not continuous data as used here. What is unclear is why the authors have chosen to model this data as incidence rate, since it is count data initially. I find the argument about continuous data to be superfluous unless the value of the incidence rate is established first. Without this argument the two-part ML approach and the authors performance metrics are still valuable and compelling. However, additional attention is brought to the incidence rate in L178 where the incidence rate is transformed with the transformation that is common for count data (log + 1), despite the fact that incidence rate will always be less than 1. 

I would like additional explanation or assurance with data exploration figures that this is not problematic. Finally, in line 520 the authors talk about quantitative models predicting count variables, which is sensible for disease data but not strictly in keeping with the focus on the incidence rate here. I would like the authors to better explain the reason incidence rate is preferable to the raw counts and point out that their analysis would work on either counts or incidence rate. It seems to me that their "hybrid model" could equally well be called a machine-learning hurdle model or similar. A hurdle model can be accurately seen as any model with two parts that do the things their models do here, although they have in the past been primarily two different generalized linear models. For example: https://hal.science/hal-03739838/ or https://link.springer.com/article/10.1007/s00607-023-01224-3

L238 I assume this should be the highest (absolute) Pearson correlation coefficient (i.e. largest positive OR negative correlation)?

Reviewer #3: good

**Results**

-Does the analysis presented match the analysis plan?

-Are the results clearly and completely presented?

-Are the figures (Tables, Images) of sufficient quality for clarity?

Reviewer #1: Line 384. The statement about model improvement with merged village datasets could have some nuance. While most models showed improvement, the performance of the RF qualitative model appears to have decreased from non-merged to merge village scales. The SVM and XGB models each had a metric show reduced performance with the merged village datasets.

Line 395. Did any village-scale temporal models have an AUC < 0.5 (worse than random chance)? The binning of AUC for equal width low, moderate, and high groups means that this cannot be ascertained from the text or Figure 3. If all villages had an AUC > 0.5, I would suggest truncating the low category at 0.5 (or whatever minimum AUC was observed) for clarity to the reader on the lower end of model performance achieved across villages.

Reviewer #2: L348 Unclear that "this resolution" refers to the week-region model, could be misinterpreted.

Reviewer #3: good

**Conclusions**

-Are the conclusions supported by the data presented?

-Are the limitations of analysis clearly described?

-Do the authors discuss how these data can be helpful to advance our understanding of the topic under study?

-Is public health relevance addressed?

Reviewer #1: Line 472. An epidemic of dengue (not just a pandemic) could also reduce the frequency of zero cases.

The Discussion would be enhanced with some discussion around the biological plausibility of identified environmental covariates and dengue incidence (Tables S4 and S5). The authors mention associations with village size and population size, but do not appear to discuss the environmental covariates.

Reviewer #2: (No Response)

Reviewer #3: good

**Editorial and Data Presentation Modifications?**

Reviewer #1: L206-207 (and following): The presentation of P-values as p<0.000 here and throughout the manuscript could appear that P-values were less than zero, which is impossible. Presenting them as p<0.001 would prevent this confusion. I defer to the journal guidelines on number of significant figures to include for these highly significant instances.

Reviewer #2: (No Response)

Reviewer #3: good

**Summary and General Comments**

Reviewer #1: Thank you for your edits and additions to clarify your manuscript. It is much improved. The clarification throughout that zero-inflated and hurdle models cannot handle continuous data is a valuable addition to the manuscript to provide clear justification and motivation for your ML method. Additionally, Fig S3 is a very useful diagram to outline the data acquisition, pre-processing, and modeling flow. Thank you for including this.

I have a few minor comments remaining (above sections and one below here).

Line 128. This explanation is true for a hurdle model, but not for zero-inflated model. In a hurdle model, all zero observations are produced by a single piece of the model (binary piece). In a zero-inflate model, zeros can arise from the binary piece of the model as well as from the count piece of the model. The structure of the hybrid model appears to align with that of a hurdle model where the quantitative piece never produces any zeros. A zero-inflated framework for the hybrid model would not necessarily be incorrect; zeros produced from the quantitative piece would relate to dengue cases being present, but not detected through passive surveillance. Clarification around hurdle vs. zero-inflated models in regards to the authors intent of their model would be appreciated.

Reviewer #2: I thank the authors for their improvements to the manuscript. Various aspects of the methods, results, and discussion were well clarified and their responses to previous reviews were careful. 

I had requested that the authors include the code used. The new version includes an example version of the code, but I don't think this matches the intention of the journal (https://journals.plos.org/plosntds/s/materials-software-and-code-sharing) that the work be reproducible, in that it only includes one algorithm and does not include the full procedure for tuning for that algorithm. The solution is to include the actual analysis code with the data in the Dryad repository.

Reviewer #3: the authors addressed the comments and questions appropriately

PLOS authors have the option to publish the peer review history of their article (what does this mean?). If published, this will include your full peer review and any attached files.

Reviewer #1: No

Reviewer #2: No

Reviewer #3: No

Figure Files:

Data Requirements:

Reproducibility:

References

---

## [Decision Letter · Decision Letter 2]

1 Oct 2024

Dear Dr. Watanabe,

We are pleased to inform you that your manuscript 'Hybrid Machine Learning Approach to Zero-Inflated Data Improves Accuracy of Dengue Prediction' has been provisionally accepted for publication in PLOS Neglected Tropical Diseases.

A few final comments from the reviewers are included below. These should be addressed with the PNTDs copyediting staff in the final version ahead of publication. The comments are minor, so we have provisionally accepted the manuscript to avoid delaying publication with another cycle of review. One reviewer reported difficulty with the Dryad data download, but we have verified that the Dryad data link now works as expected.

Before your manuscript can be formally accepted you also will need to complete some formatting changes, which you will receive in a follow up email. A member of our team will be in touch with a set of requests.

Best regards,

Christopher M. Barker

Academic Editor

Victoria Brookes

Section Editor

Reviewer's Responses to Questions

**Key Review Criteria Required for Acceptance?**

**Methods**

-Are the objectives of the study clearly articulated with a clear testable hypothesis stated?

-Is the study design appropriate to address the stated objectives?

-Is the population clearly described and appropriate for the hypothesis being tested?

-Is the sample size sufficient to ensure adequate power to address the hypothesis being tested?

-Were correct statistical analysis used to support conclusions?

-Are there concerns about ethical or regulatory requirements being met?

Reviewer #1: Line 209. Did the authors intent the use “priorly” instead of “priory”?

Reviewer #2: The authors have clarified my questions regarding methodological choices

**Results**

-Does the analysis presented match the analysis plan?

-Are the results clearly and completely presented?

-Are the figures (Tables, Images) of sufficient quality for clarity?

Reviewer #1: Line 406. Perhaps adding to this sentence to state: “The variables that showed significant Spearman’s correlation coefficient with model R2 and AUC were…” Then it is clear what these variables were related to.

Lines 433-434. Could the authors point the reader to where the 11.6% and 71.6% values come from? It is unclear which of the hybrid models the authors are referring to or if these values are averaged across models.

Reviewer #2: Good.

**Conclusions**

-Are the conclusions supported by the data presented?

-Are the limitations of analysis clearly described?

-Do the authors discuss how these data can be helpful to advance our understanding of the topic under study?

-Is public health relevance addressed?

Reviewer #1: Lines 527-542. In the discussion on limitations due to passive surveillance systems, another point the authors could include is that the proportion of symptomatic cases seeking medical care could vary spatially and temporally based on things like access to health care.

Reviewer #2: The remaining thought I had regarding the discussion and interpretation has to do with the hurdle model interpretation. The authors describe how the initial qualitative (binary) model leads to improvements in the subsequent quantitative model. What are the implications for management, or for that matter in the test set, for areas where the qualitative model predicts zero cases, but yet cases arise? These cases have not been addressed by the quantitative model at all.

**Editorial and Data Presentation Modifications?**

Reviewer #1: Line 32. The abbreviation for conditional inference forest is not needed here in the abstract.

Line 316. I believe the authors mean Text S6 since there is no Table S6.

Line 710-711. It appears that the figure legends for Figs. S11-S12 are flipped. Fig. S11 illustrates the distribution of dengue incidence while Fig. S12 is the spatiotemporal heatmap. Also, stating that the abbreviation DI means dengue incidence would clarify for readers what the x-axis represents.

Supplemental information items are not ordered sequentially based on their appearance in the main text. I defer to journal guidelines, but suggest that the authors re-number these items in sequential order to improve the logical flow of supplemental items.

Text S6. At the end of this document, S9 is listed for the code. I believe this should be Code S7 as Table S9 contains model performance metrics.

Reviewer #2: L418 - 420 Doesn’t seem consistent with itself and the following sentence

S6 text, GAMs L6, don’t understand this sentence “yet biased in linear models…”

S6 text final line refers to code in S9, I think it is in S7

**Summary and General Comments**

Reviewer #1: Thank you to the authors for the edits and improvements on their manuscript. These changes have helped clarify the manuscript results and scope. My comments are relatively minor at this point.

Reviewer #2: I thank the authors for addressing my previous concerns.

I gave up on downloading the data from Dryad. First try it was stuck on preparing download for 10 minutes, I refreshed and waited 20 minutes the second time.

PLOS authors have the option to publish the peer review history of their article (what does this mean?). If published, this will include your full peer review and any attached files.

Reviewer #1: No

Reviewer #2: No

---

## [Editor Report · Acceptance letter]

17 Oct 2024

Dear Dr. Watanabe,

We are delighted to inform you that your manuscript, "Hybrid Machine Learning Approach to Zero-Inflated Data Improves Accuracy of Dengue Prediction," has been formally accepted for publication in PLOS Neglected Tropical Diseases.

Best regards,

Shaden Kamhawi

co-Editor-in-Chief

Paul Brindley

co-Editor-in-Chief
